# A Case of Double Standard: Sex Differences in Multiple Sclerosis Risk Factors

**DOI:** 10.3390/ijms22073696

**Published:** 2021-04-02

**Authors:** Benedetta Angeloni, Rachele Bigi, Gianmarco Bellucci, Rosella Mechelli, Chiara Ballerini, Carmela Romano, Emanuele Morena, Giulia Pellicciari, Roberta Reniè, Virginia Rinaldi, Maria Chiara Buscarinu, Silvia Romano, Giovanni Ristori, Marco Salvetti

**Affiliations:** 1Centre for Experimental Neurological Therapies (CENTERS), Department of Neurosciences, Mental Health and Sensory Organs, Sapienza University of Rome, 00189 Rome, Italy; benedettamariaangeloni@gmail.com (B.A.); bellucci.1638116@studenti.uniroma1.it (G.B.); ballerini.1675668@studenti.uniroma1.it (C.B.); carmela.romano@uniroma1.it (C.R.); emanuele.morena@uniroma1.it (E.M.); giulia.pellicciari@uniroma1.it (G.P.); roberta.renie@uniroma1.it (R.R.); virginia.rinaldi@uniroma1.it (V.R.); mariachiara.buscarinu@uniroma1.it (M.C.B.); silvia.romano@uniroma1.it (S.R.); marco.salvetti@uniroma1.it (M.S.); 2San Raffaele Roma Open University, 00166 Rome, Italy; rosella.mechelli@uniroma5.it; 3Scientific Institute for Research, Hospitalization and Healthcare San Raffaele Pisana (IRCCS), 00166 Rome, Italy; 4Neuroimmunology Unit, Scientific Institute for Research, Hospitalization and Healthcare Fondazione Santa Lucia (IRCCS), 00179 Rome, Italy; 5Scientific Institute for Research, Hospitalization and Healthcare (IRCCS), Istituto Neurologico Mediterraneo (INM) Neuromed, 86077 Pozzilli, Italy

**Keywords:** multiple sclerosis, sex bias, environmental factors, genetic factors

## Abstract

Multiple sclerosis is a complex, multifactorial, dysimmune disease prevalent in women. Its etiopathogenesis is extremely intricate, since each risk factor behaves as a variable that is interconnected with others. In order to understand these interactions, sex must be considered as a determining element, either in a protective or pathological sense, and not as one of many variables. In particular, sex seems to highly influence immune response at chromosomal, epigenetic, and hormonal levels. Environmental and genetic risk factors cannot be considered without sex, since sex-based immunological differences deeply affect disease onset, course, and prognosis. Understanding the mechanisms underlying sex-based differences is necessary in order to develop a more effective and personalized therapeutic approach.

## 1. Introduction

Multiple sclerosis (MS) is a demyelinating inflammatory disease that affects the central nervous system (CNS). The chronic inflammatory process is mediated by the immune system through direct and indirect mechanisms. MS onset is extremely variable, though peak incidence is usually between 20 and 50 years. The complex interactions of the elements involved are still being studied, but it is well known that both environmental and genetic factors may contribute to disease development, making MS a multifactorial disease [1].

Women are disproportionately affected by the prevalent relapsing–remitting MS (RRMS) form, with a 3:1 preponderance [2,3,4,5]. To explain the greater female predisposition to dysimmune pathologies, the role of hormonal profile differences has been hypothesized [6]. Females experience strong hormonal changes over their lifetime. During puberty, sexual dimorphism is observed, with a change in MS prevalence from 1:1 to 3:1 [7]. A correlation between the age of menarche and disease onset has been proven by various studies [8,9]. Women develop the disease earlier, with a greater number of relapses [10] and a sharper inflammatory pattern at magnetic resonance imaging (MRI) [11], while men have a more rapid progression and a worse outcome, with severe cerebellar lesions, greater gray matter atrophy [12], and a less inflammatory but more disruptive pattern at MRI [11].

Generally speaking, women have a more effective immune response than men. A higher production of IL-21 and IL-27 (both promoting B cell maturation), a greater number of CD4+ T cells, and an important variability in the number of regulatory T (TReg) and T helper 1 (Th1) and 2 (Th2) cells according to menstrual cycle phase have been reported [5,13]. Furthermore, estradiol and its receptor (ER) show a dose-dependent influence on innate immune response, further supporting the role of sex hormones in dampening or enhancing innate immune cells [14].

During pregnancy, immunotolerance is developed in order to protect the fetus from the maternal immune system. Several hormones and cytokines (placenta-derived estriol (E3), progesterone, prolactin, leptin, glucocorticoids, alpha-fetoprotein, and insulin growth factor) increase and contribute to immune modulation. As a result, adaptive immune responses are weakened, Th1 shifts to a Th2 profile, reducing cytotoxic T cell response, and TReg cell activity increases while NK cells decrease [15,16]. Physiological pregnancy-induced immunomodulation may be the reason that some immune-mediated pathologies, such as uveitis, rheumatoid arthritis, psoriasis, and MS, remit during gestation [16,17]. According to some studies in women with MS, the relapse rate decreases up to 70% during the third trimester [18]. Consistent with the reduction in circulating hormones, disease relapses peak three months postpartum. It is possible that relapses occur due to the rapid re-assessing of prepregnancy immune activity, causing a sort of postpregnancy immune reconstitution inflammatory syndrome (PP-IRIS). It has been demonstrated that the number of relapses in the year before and during pregnancy is the most reliable factor to predict relapse occurrence during puerperium [19].

The aim of this review is to analyze the possible correlation between widely accepted MS risk factors and female sex, specifically focusing on differences from an immunological point of view and the possible role of female hormones in the interaction with MS risk factors (Figure 1).

## 2. Risk Factors

### 2.1. Environmental Factors

MS is a multifactorial disease, which means that it is caused by the interaction of multiple exogenous factors on a predisposing genetic substrate, converging with epigenetic and postgenomic regulatory events and resulting in pathology. Among the most convincing risk factors are Epstein–Barr virus (EBV) infection, cigarette smoke, lack of sun exposure, low vitamin D levels, and obesity. Even if it is possible to hypothesize a correlation with sex for each single risk factor, studies have not always been able to shed light on the sex-biased mechanisms that interact with risk/protective factors and underly the prevalence of MS in women.

#### 2.1.1. Epstein–Barr Virus

EBV is associated with a higher risk of multiple autoimmune diseases, such as dermatomyositis, systemic lupus erythematosus, rheumatoid arthritis, Sjögren syndrome, and MS [20]. Collective circumstantial evidence is compelling for MS. EBV-negative individuals show a 10-fold lower MS risk than people infected by EBV in childhood [21]. People who had infectious mononucleosis have more than twice the risk of developing MS [22,23] and, in a recent paper, it was suggested that EBV infection after adolescence is a MS risk factor [24]. Moreover, individuals with MS have much higher levels of EBNA1 antibodies, with one particular fragment (amino acids 385–420), in combination with human leucocyte antigen HLA-DRB1*15:01 status (the greatest genetic risk factor for MS), associated with a 24-fold increased risk of MS [25,26]. Accordingly, almost all EBNA1-negative individuals had serologically converted to EBNA1 antibody positive prior to MS onset [25].

EBNA1 is not the only EBV latent protein involved in MS risk: an excess of EBNA2 binding sites in regions containing single nucleotide polymorphisms (SNPs) associated to MS (and obtained from published list of genome-wide association studies (GWAS) variants [27]) was found to place individuals at a higher risk for MS and other autoimmune diseases [28,29]. Furthermore, the allelic variant 1.2 of EBNA2 has been strongly associated with MS risk [30]. Indirect evidence of the pathogenic role of EBV in MS derives from the recent success of B-cell-depleting therapies on disease course, which may be related to the fact that the virus latently infects memory B cells [31,32].

EBV represents an important index of well-being in the global population, since early infection has been correlated with poor hygienic standards and lower socioeconomic conditions [33]. The risk of contracting the infection increases with age, and multiple studies have been conducted to analyze EBV prevalence in various age groups of the general population and investigate any correlations between the onset of infection and the development of associated complications. Longitudinal studies found a significant difference in EBV prevalence between boys and girls in childhood and adolescence, showing an overall infection preponderance and higher antibody titers in females than males [2,34]. The trend persisted during adulthood, with a lower EBV prevalence in men than in women and higher EBV-specific antibody titers in seropositive women than seropositive men [35].

The above data may be correlated with the more efficient immune response promoted by estrogens compared to the immunosuppressive role of androgens. Females have a greater humoral response than males, as evidenced by higher titers of serum immunoglobulin, and a greater antibody response to various antigens after immunization [36,37]. In addition, the observation that women have a lower incidence of tumors and faster skin allograft rejection has led to the hypothesis that they also have a stronger cell-mediated immune response [38]. It has been demonstrated that estrogens directly regulate B cells by upregulating Iµ-Cµ transcripts, involved in the Ig class switch recombination process. Moreover, when enhancers bind to the immunoglobulin heavy chain locus, the estrogen receptor directly influences antibody response and sex-biased vulnerabilities to pathogens and autoimmune disease [39].

It could be speculated that the same efficiency in response to viral insults in females may underpin a higher propensity for developing immune-mediated pathologies. In the context of EBV infection, a possible relationship between higher humoral antivirus response and the propensity for developing MS deserves further direct studies regarding the pathways underpinning the impact of sex hormones on immune response to the virus.

#### 2.1.2. Cigarette Smoke

Several hypotheses have been proposed to explain the increased risk of MS among smokers, including effects on cardiovascular and immune systems, increased frequency of respiratory infections, and the neurotoxic effects of the metabolites of cigarette smoke [40]. Cigarette smoke could affect homeostasis by promoting a proinflammatory environment [41], by promoting T lymphocyte activation [42], and by disrupting the blood–brain barrier [43]. In addition, smoking seems to affect MS onset risk and prognosis (an increased risk of progression from RRMS to the secondary progressive form was demonstrated in smokers compared to nonsmokers) [40,42]. Moreover, a dose-dependent correlation with cigarette smoke exposure was demonstrated, with a 20% increased risk of developing the disease in occasional smokers and a 60% increased risk in heavy smokers. This risk factor appears to be more relevant in women [44], in whom smoke exposure seems to increase MS risk [45]. While the exact mechanisms are unknown, a correlation with sexual hormones has been hypothesized, since it was shown that oxidative DNA damage caused by smoke exposure is significantly associated with autoantibodies to oxidized DNA production in young women compared with males, and autoantibody titers were higher in females under 50 years of age compared with older female smokers [46].

A study conducted on more than 400 participants analyzed the combined effect of smoking and other well-established risk factors, such as EBV infection (considering the titer of antibodies recognizing EBV nuclear antigen 1-EBNA1) and the presence of HLA-DRB1*15:01 haplotype [47]. The results suggested a multiplicative effect, showing that never smokers with the highest anti-EBNA Ab titers had a fourfold increase in MS risk, while ever smokers with the highest anti-EBNA Ab titers had a sevenfold increase in MS risk. Although the study did not focus on sex differences, and EBV infection can be considered ubiquitous, it was demonstrated that women have a greater tendency to contract the infection and to develop higher EBV antibody titers than men.

These findings and the evidence that smoking has increased globally among women [48,49] make this risk factor a main candidate to explain a female-driven increase in MS incidence in recent decades [50].

#### 2.1.3. Lack of Sun Exposure and Low Vitamin D

Vitamin D is a steroid hormone produced by the skin from 7-dehydrocholesterol through the action of UV rays that is transformed into the active form by the kidneys and liver. Among its various biological functions, it is an important modulator of both innate and adaptive immune response [51,52]. Vitamin D can regulate the differentiation of B and T cells and reduce the activity of self-reactive T cells acting on antigen-presenting cells [53]. Its levels may fluctuate during the year and may be affected by latitude, sun exposure, and dietary intake. Reduced vitamin D levels are recognized not only as a risk factor for MS, but also as a possible predictor of disease severity and relapse rate [54].

Children, elderly, and pregnant women are at risk of vitamin D deficiency, with an increased risk of osteoporosis, infertility, cardiovascular events (especially in the postmenopausal period), and autoimmune disease onset [55,56,57,58,59]. Vitamin D deficiency occurs more frequently in postmenopausal women compared to age-matched men [55], but the multiplicity of factors influencing its levels makes it difficult to pinpoint an exact reason for this discrepancy. However, several studies show a positive correlation between circulating estrogen levels and vitamin D [60,61]. Exposure to endogenous estrogens likely induces an increase in calcidiol 25(OH)D production in the liver, but the mechanisms are still not fully understood. Human ovarian cells express both alpha-hydroxylase [62], the enzyme that converts 25(OH)D to 1.25(OH)_2_D, and the vitamin D receptor (VDR). The isoform 1.25(OH)_2_D is thought to augment aromatase activity in the human ovary, increasing estrogen production [63]. Furthermore, experiments with human ovarian cells highlighted an increase in progesterone production following vitamin D administration, and in vivo evidence showed a direct effect of vitamin D on the ovaries without affecting luteinizing hormone and follicle-stimulating hormone trajectories [64]. Conversely, it has been speculated that higher estrogen levels (both exogenous and endogenous) may increase vitamin D binding protein (VDBP) levels [60], which in turn extend the half-life of vitamin D. To explore the association between genetic susceptibility to MS and vitamin D, the SNPs of 12 vitamin D-related genes have also been studied, but the results, often conflicting, have not been able to clarify this point [65].

The studies reported so far seem to suggest that estrogens positively affect vitamin D levels, making it unlikely that the sex bias in MS is directly mediated by vitamin D deficit.

#### 2.1.4. Obesity

Obesity is a multifactorial condition, defined by body mass index (BMI) > 30 kg/m^2^, which predisposes individuals to various comorbidities, including diabetes, cardiovascular pathologies, and neoplasms [66]. During adolescence, obesity prevails among males [67], while in adulthood women are the most affected [68]. Adipose tissue is closely linked to the activity of sex hormones in many aspects, being influenced by fat distribution and its activity. Differences become more obvious during adolescence, consistent with the onset of hormonal profile differences [69]. Obesity and overweight are linked to early menarche [70] and earlier pubertal development stages in females [71]. Obesity during adolescence has been associated with an increased risk of developing MS [72] due to its correlation with both a chronic inflammation status (especially through the production of cytokines, such as leptin [73], which influences immune response), and with low levels of vitamin D [72]. Throughout life, androgens constantly exert a suppressive effect on adiponectin and leptin. Testosterone affects glucose metabolism by increasing its uptake. During old age, when testosterone levels drop, visceral adiposity increases [74]. Leptin plays a key role in the induction and maintenance of an effective proinflammatory immune response in the CNS by modulating pathogenic CD4+ cells, and mice lacking circulating leptin do not develop experimental autoimmune encephalomyelitis (EAE) after immunization [75]. Females have higher levels of leptin, especially during puberty due to the action of estrogen [76], and because subcutaneous fat produces more leptin than visceral fat [77]. This evidence suggests a prevalent role of female obesity in facilitating MS development.

Conversely, longitudinal epidemiological studies showed that the correlation between obesity and MS susceptibility (double risk of developing MS for a BMI > 27 kg/m) exists without significant differences between males and females^2^ [78]. How obesity acts as an independent risk factor, irrespective of sex, is unclear, but a possible explanation could be linked to the different properties of adipose tissue in the two sexes. Men have a higher tendency to store fat in the abdominal and visceral areas (android distribution), while women tend to present subcutaneous and gluteal-femoral fat (gynoid distribution) [79]. Although men produce less leptin and adiponectin, it has been demonstrated that visceral fat is associated with higher proinflammatory cytokine production [80], suggesting that sex-specific obesity-related risk factors favor MS development, with possible immune effector differences in females as compared with males.

### 2.2. Genetic Factors

Male and female differences in immune response regulation depend not only on sex hormones, but also on sex-biased genetic differences. In particular, sex-chromosome complement has been identified as a determinant of the female bias in autoimmune disease prevalence [81]. Changes in X-inactivation mechanisms could explain the higher gene dosage differences in females (XX) than males (XY).

It has recently been demonstrated that due to escape from X chromosome inactivation (XCI), females express higher levels of TLR7 in B cells, enhancing the IgG class switch [82]. Similarly, other genes located on the X chromosome, such as those encoding for CD40-ligand and Bruton’s tyrosine kinase (BTK), can escape XCI and may promote the induction and maintenance of antibody responses in women [83]. The most sexually dimorphic expressed gene in human and murine CD4+ T cells, due to XCI escape, was *Kdm6a*, which codes for the histone demethylase that suppresses trimethylation on H3K27 (H3K27me3) and promotes chromatin transcription. The authors found a decrease in neuropathologic severity after *Kdm6a* deletion in CD4+ T cells of EAE mice. Since *Kdm6a* is expressed in both female alleles in mice and humans, it may plausibly contribute to the genetic factors responsible for increased MS risk in women [84]. Another study confirmed the importance of the sex chromosome complement in transgenic SJL mice: both EAE and pristane-induced lupus showed a more severe course, which was associated with the XX complement as compared with the XY complement [85]. It has also been suggested that the Y chromosome contains polymorphic genes that might reduce EAE susceptibility [86], corroborating the finding that both hormones and sex chromosomes differentially affect autoimmune response, and that the sex chromosome complement directly affects autoantigen-specific immune response [87].

The fact that the X chromosome can be inherited from either the mother or father has also been investigated as a determinant of sex-biased MS risk. The interleukin-18 (*IL-18*) gene (which encodes for a cytokine modulating brain homeostasis and neuroinflammation [88] and is linked to MS [89]) is more likely to be expressed from the maternal allele in the preoptic area of the hypothalamus and medial prefrontal cortex of females, suggesting the presence of a parent-driven epigenetic mechanism influencing gene expression in females, but not males [90]. This phenomenon is part of the “maternal parent-of-origin effect”, which has been repeatedly observed in various diseases, including juvenile idiopathic arthritis [91], myotonic dystrophy [92], and MS [93,94,95]. Twin studies conducted by Sadovnik et al. in MS populations suggested a key role of the maternal parent-of-origin effect in influencing the age of disease onset [96]. Another group found increased methylation in the paternal X (Xp) rather than the maternal X (Xm) in the upstream region of the *Foxp3* enhancer, with consequent suppression of the *Foxp3* gene in XmXp (prevalent suppressive effect of *Foxp3* expression by the paternal allele) in comparison with XmY (lack of suppressive effect of *Foxp3* expression by the maternal allele), which led to decreased *Foxp3* expression and increased immune response in EAE female mice [97].

The HLA-DRB1*15:01 haplotype is considered the major individual genetic risk factor for MS, while other histocompatibility molecules, such as those encoded by the HLA-A2 haplotype, were found to be protective [98]. Several studies consider the HLA-DR2 haplotype as an interactor with other risk/protective factors for disease development (see above). Literature concerning the relationship between HLA molecules and sex bias is relatively scarce. An Iranian study investigated the role of non-HLA genes in HLA-DRB1*15:01-negative patients to explain the sex bias in MS. In particular, the authors studied the gene coding for a long non-coding RNA expressed in dendritic cells (*lnc-DC*), since its downregulation resulted in decreased CD4+ T cell proliferation and cytokine production. They found increased expression of *lnc-DC* in female HLA-DRB1*15:01-negative patients, suggesting a role of this immunological component in MS sex bias [99].

Genome-wide association studies (GWAS) have recently made a substantial contribution to etiologic studies on MS, showing a complex genetic architecture with many common SNPs and a few rare variants that predispose individuals to the disease [100]. International multicentric studies of tens of thousands of cases and controls identified approximately 200 autosomal variants other than HLA capable of modifying MS risk/protection. The majority of detected SNPs were found in regulatory regions associated with immune response, and frequently shared with other autoimmune diseases. Future reworking of GWAS data will allow a better understanding of the role of MS-associated variants as determinants of disease sex bias. In the meantime, direct evidence from the last GWAS found a susceptibility locus on the X chromosome (rs2807267), which supports the importance of genetic factors in sex-related bias in MS [100]. Evidence of a susceptible X chromosome locus is of interest and will allow further and more targeted experiments on genetic contribution to sex bias.

## 3. Conclusions

The female sex bias in MS has been well known for a long time, but the exact underlying processes are still unclear. Understanding the interactions between several environmental factors and multiple genetic sex-biased mechanisms still requires further study. The complex role of genetic factors in the pathogenesis and progression of multifactorial human diseases, such as MS, has driven researchers towards new approaches. In particular, the aforementioned GWAS analyses seem promising. A recent report from the Genotype-Tissue Expression (GTEx) project highlighted the presence of mutations in sex-specific genes and the role of sex-differential human gene expression in human disease onset [101]. Furthermore, the mapping of sex-biased expression quantitative trait loci (sb-eQTL) has shown sex differences in the gene expression of tissues analyzed in the GTEx project [102]. Concerning MS, one important point highlighted by the last GWAS regards the role of microglia in MS development [100], which may contribute to sex bias. In fact, increasing evidence suggests that microglia differ in number and morphology in almost every CNS area and play a central role in sexual differentiation during brain development, influencing processes such as cell proliferation, synaptic connectivity, and cellular physiology [103]. These distinct differences may lead to a sui generis sex-biased microglial inflammatory response in the brain, influencing damage repair in a sex-dependent fashion [104]. Consistent with this hypothesis, a rodent study on an experimental model of MS suggested that estrogens modulate symptom severity by inhibiting microglial activation [105].

Further reworking of GWAS data and other high-throughput approaches focused on sex chromosomes and hormone-related loci [102] may contribute to better understanding the genetic basis of MS sex bias. Results from these studies will contribute to the implementation of new therapeutic approaches. In particular, the development of personalized treatment with an optimal therapeutic index and high preventive effects must take into account sex bias as a main determinant in patient stratification during daily clinical practice.

## Figures and Tables

**Figure 1 ijms-22-03696-f001:**
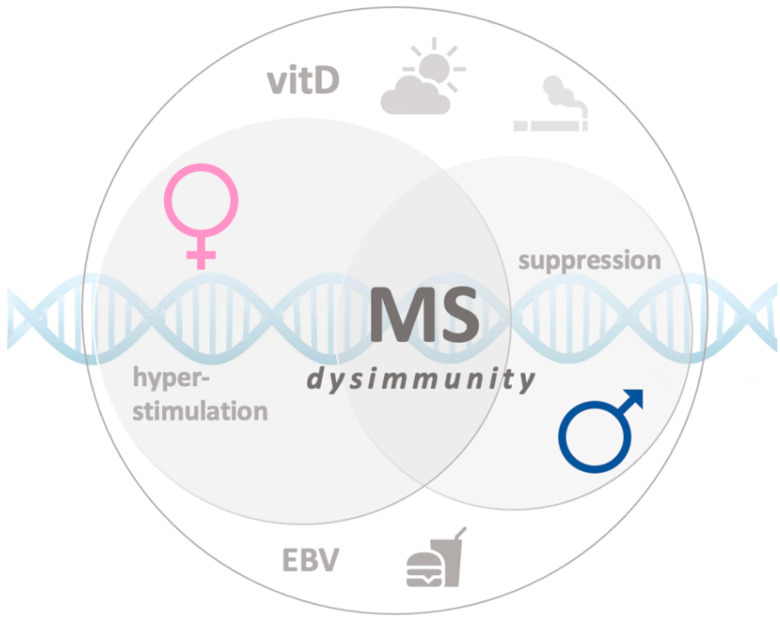
Schematic representation of the environmental and genetic factors involved in multiple sclerosis (MS) sex bias. (EBV: Epstein-Barr virus)

## Data Availability

Not applicable.

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
