# Peer review of "A Case of Double Standard: Sex Differences in Multiple Sclerosis Risk Factors"

_ijms, 2021, doi:10.3390/ijms22073696_

Round 1

Reviewer 1 Report

The manuscript is very interesting and well-written.

I have some minor suggestions to improve its quality.

I suggest to the Authors to add a figure and a table in order to better capture the attention of the readers. 

Please, consider to cite the following articles: Biochem Genet. 2021 Feb;59(1):1-30. doi: 10.1007/s10528-020-10010-1.

Medicina (Kaunas). 2019 Jul 5;55(7):341. doi: 10.3390/medicina55070341.

Cent Eur J Immunol. 2018;43(3):331-334. doi: 10.5114/ceji.2018.80053. 

Front Neurol  2020 Jul 3;11:616.  doi: 10.3389/fneur.2020.00616.

Author Response

Response to Reviewer 1 Comments

Point 1: I suggest to the Authors to add a figure and a table in order to better capture the attention of the readers

Response 1: We thank for the suggestion. We have already inserted a figure, to schematize the aim of this review. We didn’t include in the text data lists to make a table.

Point 2: Please, consider to cite the following articles:

Biochem Genet. 2021 Feb;59(1):1-30. doi: 10.1007/s10528-020-10010-1.

Medicina (Kaunas). 2019 Jul 5;55(7):341. doi: 10.3390/medicina55070341.

Cent Eur J Immunol. 2018;43(3):331-334. doi: 10.5114/ceji.2018.80053. 

Front Neurol  2020 Jul 3;11:616.  doi: 10.3389/fneur.2020.00616.

Response 2: We thank for the comment and we added all the suggested reference:

Biochem Genet. 2021 Feb;59(1):1-30. doi: 10.1007/s10528-020-10010-1 is now the reference 64

Medicina (Kaunas). 2019 Jul 5;55(7):341. doi: 10.3390/medicina55070341 is now the reference 57

Cent Eur J Immunol. 2018;43(3):331-334. doi: 10.5114/ceji.2018.80053 is now the reference 50

Front Neurol  2020 Jul 3;11:616.  doi: 10.3389/fneur.2020.00616 is now the reference 6

Reviewer 2 Report

The aim of this review is to analyze the possible correlation between widely accepted MS risk factors and female sex, specifically focusing on differences from an immunological point of view and the possible role of female hormones in the interaction with MS risk factors.

I'm a statistician working on MS.

I think that this review is well written, very clear and interesting and the message is important.

Author Response

No response or corrections were request by this reviewer. We thank for the the comments.

Reviewer 3 Report

This manuscript is bringing up the differences between the sexes as an important aspect of MS. Environmental factors like EBV, VitD, Smoking and obesity is discussed, as well as genetic factors. Indeed the sex differences seem to drive many aspects of immunity and infection and I agree with the authors that we have to be better in integrating these aspects better in our understanding of the mechanism of MS. In addition, the fluctuating hormonal cycles and age changes in women also have to be considered. 

A few major and minor suggestions and comments can be given to improve the manuscript. 

Major:

The words gender and sex are used interchangeably in the manuscript, although sex usually refers to the biological aspect and gender more to the cultural. Please make a sentence on how you define it and then keep to this definition. 

An important finding regarding age and EBV as risk factor came out of the Sundström's group recently (Biström et al Eur J Imm 2020), about EBV being a risk factor only in age over 18 yrs. 

The vitD effect is also influencing T cells directly, in humans but not in mice, which might have a more pronounced impact on MS. Therefore the increased levels of estrogen giving increased levels of vitD could still have an effect on MS, if it leads to more active autoreactive T cells. See Marina Rode von Essen et al, "Vitamin D controls T cell antigen receptor signaling and activation of human T cells" in Nature Imm 2009, doi:10.1038/ni.1851. 

Minor:

HLA-DRB1*15:01 is written wrong the first time it appear in line 154 (but not the second). 

Line 99-100 on EBNA-2 binding to SNP is unclear. What do you mean here? Please clarify. 

Line 124: "up-regulating transcripts" is not very precise. What transcripts and how would that make an impact?

Line 176: "compared to men", were these age matched? 

Line 267-286 about the protective effect of HLA-A2 and reference 92 which is a review. Please refer to the original publication of this finding. 

Line 271 regarding the differences in Iran, it does not say in what direction. Please add the actual frequencies and the significance of it. Has this been replicated by several studies? 

Author Response

Response to Reviewer 3 Comments

Point 1: The words gender and sex are used interchangeably in the manuscript, although sex usually refers to the biological aspect and gender more to the cultural. Please make a sentence on how you define it and then keep to this definition. 

Response 1: We thank for this comment and, since in this review we are describing biological aspects, we decided to use just the word ‘sex’.

Point 2: An important finding regarding age and EBV as risk factor came out of the Sundström's group recently (Biström et al Eur J Imm 2020), about EBV being a risk factor only in age over 18 yrs.

Response 2: We changed the sentence in: “People who had infectious mononucleosis have more than twice the risk of developing MS[22, 23] and in a recent paper it has been suggested that EBV infection after adolescence is a MS risk factor[24].”, with the suggested reference.

Point 3: The vitD effect is also influencing T cells directly, in humans but not in mice, which might have a more pronounced impact on MS. Therefore the increased levels of estrogen giving increased levels of vitD could still have an effect on MS, if it leads to more active autoreactive T cells. See Marina Rode von Essen et al, "Vitamin D controls T cell antigen receptor signaling and activation of human T cells" in Nature Imm 2009, doi:10.1038/ni.1851. 

Response 3: We thank for this comment, but the following sentence is reported in the conclusions of the cited paper: ‘Given that T cells are capable of explosive proliferation, the lag phase imposed by the vitamin D-VDR ‘prelude’ may diminish the risk of unwanted immunopathology’. Based on this, we are not sure this specific pathway could have effect on MS leading more autoreactive T cells.

Point 4: HLA-DRB1*15:01 is written wrong the first time it appear in line 154 (but not the second). 

Response 4: We apologize and we corrected the error.

Point 5: Line 99-100 on EBNA-2 binding to SNP is unclear. What do you mean here? Please clarify. 

Response 5: We changed the sentence in: “EBNA1 is not the only EBV latent protein involved in MS risk: an excess of EBNA2 binding sites in regions containing single nucleotide polymorphisms (SNPs) associated to MS (and obtained from published list of GWAS variants[27]) has been found to place individuals at a higher risk for MS and other autoimmune diseases[28, 29].”

Point 6: Line 124: "up-regulating transcripts" is not very precise. What transcripts and how would that make an impact?

Response 6: We changed the sentence in: “It has been demonstrated that estrogens directly regulate B cells by up-regulating Iµ-Cµ transcripts, involved in the Ig class switch recombination process.”

Point 7: Line 176: "compared to men", were these age matched? 

Response 7: We thank for the comment and we change the sentence in “Vitamin D deficiency occurs more frequently in postmenopausal woman compared to age matched men…” (line 175-176).

Point 8: Line 267-286 about the protective effect of HLA-A2 and reference 92 which is a review. Please refer to the original publication of this finding. 

Response 8: Actually the reference cited by the reviewer (Stürner, K.H., et al., Is multiple sclerosis progression associated with the HLA-DR15 haplotype? Mult Scler J Exp Transl Clin, 2019. 5(4): p. 2055217319894615.) is a research article.

Point 9: Line 271 regarding the differences in Iran, it does not say in what direction. Please add the actual frequencies and the significance of it. Has this been replicated by several studies? 

Response 9: We thank for the comment. We erased the sentence ‘A study in Iranian MS patients demonstrated that this allele had a different frequency in males and females and was associated with increased MS risk in female’, since was added in the text for an error. The Iranian study we were referring to was the one described immediately after, corresponding to the reference.